# The Management of Pregnancy Complicated with the Previable Preterm and Preterm Premature Rupture of the Membranes: What about a Limit of Neonatal Viability?—A Review

**DOI:** 10.3390/diagnostics12082025

**Published:** 2022-08-22

**Authors:** Stepan Feduniw, Zuzanna Gaca, Olga Malinowska, Weronika Brunets, Magdalena Zgliczyńska, Marta Włodarczyk, Anna Wójcikiewicz, Michał Ciebiera

**Affiliations:** 1Department of Reproductive Health, Centre of Postgraduate Medical Education, 01-004 Warsaw, Poland; 2Czerniakowski Hospital, 00-739 Warsaw, Poland; 3Faculty of Medicine, Medical University of Bialystok, 15-089 Bialystok, Poland; 4Faculty of Medicine, Lazarski University, 02-662 Warsaw, Poland; 5Department of Obstetrics, Perinatology and Neonatology, Centre of Postgraduate Medical Education, 01-809 Warsaw, Poland; 6Department of Biochemistry and Pharmacogenomics, Faculty of Pharmacy, Medical University of Warsaw, 02-097 Warsaw, Poland; 7Centre for Preclinical Research, Medical University of Warsaw, 02-097 Warsaw, Poland; 8Second Department of Obstetrics and Gynecology, Centre of Postgraduate Medical Education, 01-809 Warsaw, Poland

**Keywords:** previable PPROM, preterm rupture of membranes, PTB, preterm delivery, antibiotics

## Abstract

Preterm premature rupture of the membranes (PPROM) at the limit of viability is associated with low neonatal survival rates and a high rate of neonatal complications in survivors. It carries a major risk of maternal morbidity and mortality. The limit of viability can be defined as the earliest stage of fetal maturity when a fetus has a reasonable chance, although not a high likelihood, for extra-uterine survival. The study reviews available data on preventing preterm delivery caused by the previable PPROM, pregnancy latency, therapeutic options including the use of antibiotics and steroids, neonatal outcomes, and future directions and opportunities.

## 1. Introduction

Preterm premature rupture of membranes (PPROM) is diagnosed when the rupture occurs before 37 weeks of gestation and may be associated with preterm labor if uterine contractions also develop before term [1]. According to different sources, delivery is called a miscarriage when it occurs before 22–24 weeks of gestation [2]. Preterm rupture of membranes at this time is called previable PPROM, as the survival chances are minimal when delivery occurs. The cut-off between preterm labor and miscarriage varies around the world between 22 and 24 weeks of gestation because of poor neonatal outcomes after delivery before the 24th week of gestation and different levels of neonatal care worldwide. Perinatal mortality and severe morbidity decrease significantly after the 28th week of gestation.

PPROM complicates up to 3% of pregnancies [1,2,3,4]. The risk factors include a history of PPROM, (13.5%, vs. patients without prior PPROM 4.1%, (RR = 3.3, 95% CI: 2.1–5.2) [5]) and bacterial vaginosis (a pregnancy complication often leading to intrauterine infections, including chorioamnionitis and, as a result, preterm labor) [6,7]. Bleeding in the first trimester of pregnancy is also associated with higher PPROM incidence [8,9]. In such pregnancies, preterm labor occurs in 2.7–4.8% of cases (OR = 1.83, 95% CI: 1.7–2.0) [9]. Socio-economic factors, including cigarette smoking, also predispose to PPROM [10,11]. The abovementioned and other known risk factors are summarized in Table 1 [12].

The study aimed to review the data available in the literature on the prevention of preterm delivery caused by the PPROM and therapeutic options in PPROM newborns at the limit of viability before 24 weeks of gestation.

## 2. Clinical Issues

### 2.1. Pregnancy Latency

The duration of pregnancy is a critical prognostic factor for the newborn and is related to the duration of PROM and oligohydramnios [13,14]. Many women diagnosed with PPROM will deliver within one week of this event. In the study of 239 group B. *Streptococcus*-negative patients with PPROM at 24–32 weeks of gestation, the median continuation of pregnancy was 6.1 days despite prophylactic antibiotic treatment. Overall, 56% of patients gave birth within seven days, 76% within 14 days, and 86% within 21 days [15,16,17].

Planned early birth vs. prolongation management in patients with previable PPROM was associated with an increased occurrence of respiratory distress syndrome (RDS), higher rate of ventilation support, endometritis, and increased caesarean section rates, with a decreased rate of chorioamnionitis. For women with previable PPROM and without indications for immediate delivery, careful monitoring could be associated with lower mortality and better perinatal outcomes [18].

According to some data, cesarean delivery could be advantageous for extremely premature fetuses, according to several studies. However, there are also reports of worse postoperative adaptation of premature fetuses [19,20]. The present scientific information does not support the recommendation for cesarean delivery to improve survival or reduce morbidity in severely preterm fetuses [21].

As pregnancy with ruptured membranes continues, the risk of an intrauterine infection increases. Clinical inflammation of the membranes is defined as the presence of an elevated body temperature (>37.8 degrees C) and at least two additional factors, such as leukocytosis (>15 G/L), maternal tachycardia (>100 beats/min), fetal tachycardia (>160 heartbeats/min), uterine muscle tenderness, and abnormal and even foul-smelling amniotic fluid [22,23]. Nevertheless, the diagnosis is challenging because patients with inflammation of the membranes can be mildly symptomatic or asymptomatic. However, an infection of the membranes could lead to neonatal and even maternal death. In the physiological situation of the preserved amniotic membranes, the intra-developing fetus has no contact with the microorganisms of the external environment. Since the fetus does not have a sufficiently developed immune system, an infection can lead to intrauterine death in the short term. Dead fetal tissue and infected endometrium after fetal extraction commonly cause maternal sepsis and death.

Research shows that expectant management in women with previable PPROM at <24 weeks of gestation was associated with higher rates of maternal morbidity (aOR = 3.47, 95% CI: 1.52–7.93), chorioamnionitis (OR = 3.1, 95% CI: 2.03–8.26), and postpartum hemorrhage (OR = 2.44, 95% CI: 1.13–5.26) as compared to the termination of pregnancy [24].

LeMoine et al. showed that after previable PPROM, which occurred at 18–22 weeks of gestation, the neonatal survival rate was only 28%. Antibiotic usage in the study was associated with increased pregnancy latency. The pregnancy duration and weeks of gestation of PPROM were major predictors of neonatal mortality [25].

González-Mesa et al. assessed previable PPROM cases before 28 weeks of gestation over the period of 20 years between 2000–2020. The study showed that pregnancy duration after PROM was around 14 days, and latency increased over the years. With 53% of caesarean deliveries, the mortality rate was around 26.5% and decreased over the analyzed period (*p* < 0.05) [26].

An essential component of managing previable PPROM pregnancies is the time of delivery. This is undoubtedly one of the greatest clinical dilemmas. While extreme prematurity is associated with high neonatal mortality, prolonging pregnancy increases the risk of developing an intrauterine infection. It seems reasonable to schedule an elective delivery of pregnancies complicated by previable PPROM after reaching 32–34 weeks of gestation. At this stage, more than 98% of neonates will survive without harm to health [27,28]. The long duration of pregnancy after previable PPROM is an additional risk factor for the postpartum development of the child, while oligo-/ahydramnion leads to inappropriate pulmonary development and is secondary to increased cardiac overload [13,14]. As a result, these changes may be lethal for the newborn. If continuing the pregnancy might pose a risk to the life of the mother or fetus, the pregnancy should be delivered earlier. When choosing the delivery/termination date, obstetricians should consider the following factors: clinical diagnosis of inflammation of the membranes, the occurrence of maternal complications (e.g., impending eclampsia, HELLP syndrome), the occurrence of fetal indications (e.g., impending intrauterine asphyxia, bleeding suggesting premature separation of the properly seated placenta, umbilical cord prolapse), and elevated inflammatory markers.

### 2.2. Antibiotics

Women with PROM and PPROM are at an increased risk of intrauterine infection. Therefore, antibiotics are widely used in this indication. The 2013 Cochrane review showed that the prophylactic use of antibiotics is associated with pregnancy latency and lower short-term neonatal morbidity but without a significant reduction in perinatal mortality or long-term outcomes. No specific antibiotic therapy was recommended, but concomitant administration of amoxicillin with clavulanic acid was contraindicated due to the increased risk of necrotizing enterocolitis (NEC) in neonates [29]. The Cochrane review from 2017 shows remarkable differences in intrauterine infections in women without receiving prophylactic antibiotics [18]. Therefore, routine usage of antibiotic prophylactic is suggested. Table 2 contains a brief of recommendations of selected scientific societies regarding antibiotic therapy in PPROM. They all warn against the use of clavulanic acid due to the proven increased risk of neonatal NEC [30,31,32,33].

A randomized, double-blind, placebo-controlled trial from the National Institute of Child Health and Human Development Maternal–Fetal Medicine compared a seven-day course of sequential antibiotic therapy with 2 g ampicillin intravenously (i.v.) every six hours with erythromycin 250 mg every six hours for 48 h and then amoxicillin orally (p.o.) (250 mg every eight hours) with erythromycin 333 mg.

The ORACLE I study was performed on 4826 women and compared the following: amoxicillin with clavulanic acid with erythromycin, amoxicillin alone with placebo, erythromycin with placebo, and two placebo tablets. The drugs were taken for ten days or until delivery. Benefits were demonstrated in the group with erythromycin compared to placebo [34], and amoxicillin with clavulanic acid also showed some benefits. The National Institute for Health and Care Excellence recommendations from 2015, updated in 2019, were originally prepared based on ORACLE I study [32]. The Royal College of Obstetricians and Gynaecologists (RCOG) recommended prophylactic antibiotic therapy with 250 mg erythromycin p.o. four times a day for ten days [32]. The Society of Obstetricians and Gynaecologists of Canada (SOGC) recommendations are based on the above research and recommend prophylactic antibiotics when fetal lung maturity is not documented [33]. ACOG recommends antibiotic prophylaxis as a two-day therapy with ampicillin i.v. and erythromycin followed by five-day oral prophylaxis with amoxicillin and erythromycin. The therapy seems to prolong pregnancy and decrease short-term neonatal complications [35].

Since erythromycin is unavailable in several countries, an alternative antibiotic therapy regimen was needed. Chang et al. compared erythromycin i.v. with p.o. clarithromycin, and both were used in combination with the first generation of cephalosporins for seven days or until delivery [36]. Clarithromycin was found to be more effective in reducing the incidence of chorioamnionitis, intraventricular bleeding, and bronchopulmonary dysplasia in newborns. The higher efficacy of clarithromycin was probably due to its higher transfer rate across the placental barrier (compared to the minimal for azithromycin and erythromycin) [37]. This transfer depends on the week of gestation and is extremely low in the first trimester. As confirmed by numerous studies [38,39,40], macrolides are considered safe in pregnancy and have an additional immunomodulatory effect that may reduce the production of pro-inflammatory cytokines IL-8 and TNF alfa [41].

Pierson et al. compared ampicillin and erythromycin vs. ampicillin and azithromycin regimens. They showed that a single-dose azithromycin regimen is as effective and safe as the erythromycin regimens while being cheaper and better-tolerated by patients [42]. Another study compared azithromycin 1000 mg p.o. vs. azithromycin 500 mg p.o. followed by azithromycin 250 mg p.o. for four days, vs. azithromycin 500 mg i.v. for two days followed by azithromycin 500 mg p.o. for five days, and erythromycin i.v. for two days followed by erythromycin p.o. for five days. This study showed that azithromycin was as effective as erythromycin in all groups apart from the five-day regimen where respiratory distress syndrome occurred more often than in other groups [37]. However, there was no difference in the pregnancy latency in PPROM patients between the single 1 g dose of oral azithromycin versus erythromycin and ampicillin/amoxicillin.

Information about long-term neonatal outcomes after antibiotic prophylaxis is limited. The ORACLE study shows better functional development in children with intrauterine antibiotic administration after PPROM than those without antibiotic prophylaxis at seven years old. However, no differences in neurodevelopment were observed in the same study, where the children were followed-up until eleven years of age [43].

The most common pathogens isolated from a vaginal area of women with PROM include *Streptococcus* spp., *Staphylococcus aureus*, *Escherichia coli*, *Proteus mirabilis*, *Bacteroides* spp. and *Klebsiella pneumoniae* [44,45,46,47]. In many clinical centers, the drug of choice is ampicillin with sulbactam 1.5 g i.v. every six hours until delivery or up to seven days in combination with clarithromycin 0.5 g i.v. or p.o. every twelve hours until the end of labor or for a maximum of seven days. As much as 97.6% of the above bacteria were sensitive to the above combination of antibiotics [48]. In another study examining 1133 placental or amniotic samples, 65% contained Gram-negative *Enterobacteriaceae.* Only 38% of *Escherichia coli* and *Klebsiella* sp. were sensitive to ampicillin. Therefore, the prophylactic usage of antibiotics in women with PPROM should target Gram-negative *Enterobacteriaceae* [49].

Patients with PROM and PPROM should be monitored for group B *Streptococcus* (GBS) in the anogenital area, as this bacterium is the leading cause of septicemia, meningitis, and pneumonia in neonates. In patients with a positive swab test, preventive bacterial eradication with ampicillin with sulbactam 3 g i.v. every six hours until delivery or up to seven days seems necessary [31,50].

Meta-analysis of 20 studies conducted by Chatzakis et al. showed a reduced incidence of chorioamnionitis using clindamycin with gentamycin (RR = 0.19, 95% CI: 0.05–0.83), penicillin (RR = 0.31, 95% CI: 0.16–0.6), ampicillin/sulbactam with amoxicillin/clavulanic acid (RR = 0.32, 95% CI: 0.12–0.92), ampicillin (RR = 0.52, 95% CI: 0.34–0.81), and erythromycin with ampicillin (RR = 0.71, 95% CI: 0.55–0.92). Only erythromycin was shown to reduce the incidence of neonatal sepsis (RR = 0.74, 95% CI: 0.56–0.97). Clindamycin with gentamycin (RR = 0.32, 95% CI: 0.11–0.89) and erythromycin combination with ampicillin or amoxicillin (RR = 0.83, 95% CI: 0.69–0.99) lowered the incidence of respiratory distress syndrome. Ampicillin and penicillin have been shown to prevent grade-3/4 intraventricular hemorrhage (RR = 0.42, 95% CI: 0.20–0.92 and RR = 0.49, 95% CI: 0.25–0.96, respectively). However, no single antibiotic was clearly superior to others, and none reduced the rate of neonatal death, perinatal death, or NEC [51].

According to available data, there are ongoing clinical trials. In NCT01503606, cefazolin 1.0 g i.v. every 12 h vs. clarithromycin 500 mg p.o. two times daily for one week or until delivery is being assessed in the III phase of the clinical trial to evaluate neonatal composite morbidity.

The IV phase of a clinical trial (NCT05345457) assesses pregnant women with previable PPROM between 18 0/7 and 22 6/7 receiving azithromycin 500 mg single-dose oral, followed with 250 mg once daily for 4 additional days vs. those receiving amoxicillin 500 mg p.o. three times daily for 7 days after discharge. A primary outcome is a delivery within 28 days.

### 2.3. Tocolysis

Tocolysis is a medical procedure carried out with the use of medications with the purpose of delaying the delivery of a fetus in women presenting preterm contractions. The only proven indication is to save steroid administration time and to transfer of women in labor between hospitals [52].

The Cochrane review from 2014 did not show any advantages of tocolysis in maternal and infant morbidity and mortality in PPROM cases, and it also showed an increase in maternal chorioamnionitis. However, it should be noted that no antibiotics or steroids were used [53]. Conversely, Chackowicz et al. showed that tocolysis reduces the risk of neonatal septic death at seven days (administered 24–27 weeks of gestation) (OR = 0.44, 95% CI: 0.22–0.88) [54]. Another study found no reduction in neonatal mortality or morbidity using tocolysis in women without PPROM [55]. A group of patients using nifedipine had a lower incidence of NEC, intraventricular hemorrhage, and respiratory distress syndrome. Atosiban and nifedipine seem useful for tocolysis in PPROM patients, as they effectively delay labor and have good maternal tolerance. Betamimetics were inferior due to the atosiban and nifedipine to their cardiovascular adverse effects [55]. The III phase of a clinical trial (NCT03976063) assesses perinatal morbidity and prolongation of gestation after nifedipine vs. placebo administration in previable PPROM pregnancies.

In the past, magnesium sulphate was frequently used to induce tocolysis. However, it is used neither as a tocolytic agent in any pregnancy nor in patients with PPROM anymore, as it does not seem to improve maternal or neonatal outcomes [56], which was also proven in the Cochrane analysis 2014 [57].

### 2.4. Steroids

A course of steroid therapy with betamethasone 12 mg given twice 24 h apart or Dexamethasone 6 mg four doses in 12 h apart i.m. should be given before 35 + 6 weeks of pregnancy to promote maturation of fetal lungs [30,58,59]. Steroid therapy is only effective in preventing respiratory disorders in newborns born within 14 days after dose administration, and repeated administration of steroids during pregnancy does not improve neonatal outcomes. For this reason, steroid therapy should be initiated only in those patients who are likely to give birth within two weeks of the start of the therapy. The use of prenatal steroids may be associated with maternal side effects and result in generalized bacterial infection, tuberculosis, and viral infections (shingles, Herpes); uncontrolled metabolic diseases, including diabetes and unresponsive hypertension; or active gastric ulcer and duodenum [60,61]. However, the risk of generalized fetal and maternal infection occurrence should not decide about initiating the prophylactic intervention, as the advantages of earlier pulmonary development were proven and are much higher than the side effects of steroids administration [30,58].

### 2.5. Amnioinfusion and Other Therapies

Amnioinfusion refers to the instillation of fluid into the amniotic cavity. Augmenting amniotic fluid volume may decrease problems associated with a severe reduction or absence of amniotic fluid in PROM, thus improving neonatal outcomes. Results for amnioinfusion are limited and based on a few retrospective analyses [62,63]. The III phase of the clinical trial AmnionFlush (NCT04696003) is ongoing. In patients with PPROM between 20/0 and 26/0 weeks of gestation, amnioinfusion was performed to decrease the incidence of bronchopulmonary dysplasia, intraventricular hemorrhage (IVH 3–4) and NEC.

An experimental study on a small group of women compared amniotic fluid-replacement port system at 14–26 weeks of gestation for long-term amnioinfusion. While the results were promising [63] in terms of improving neonatal outcomes, due to methodological problems and a small study group, there is insufficient evidence to establish that the procedure reduces neonatal death, neonatal sepsis, and pulmonary and puerperal sepsis. Hence, further investigations should be performed.

There are also studies suggesting that supplementing additional vitamins and minerals could improve neonatal outcomes [64,65]. Additional supplementation could be taken into consideration by PPROM management but not as the only therapy.

### 2.6. Medical Care of Pregnant Women and Fetuses

A Cochrane review from 2014 did not find enough evidence to establish the superiority of hospital and home care for patients with PPROM. Nevertheless, outpatient clinic control of the pregnancy is required [66]. In the study of Garabedian et al., patients who experienced previable PPROM at 24 and 35 weeks of gestation and were managed at home had longer pregnancy latency and no differences in perinatal outcomes to hospital management [67]. Other researchers report similar results [68,69]. The ongoing clinical trial NCT04413019 assesses the advantages of outpatient monitoring of fetuses after PPROM between 28 and 36 weeks of gestation.

However, it should be noted that previable PPROM patients should undergo regular body temperature monitoring and laboratory control tests (e.g., leukocytosis and CRP) to detect any ongoing inflammatory processes. Since procalcitonin and serum interleukin 6 levels tests are sensitive but expensive, their use might be limited [70,71]. To assess the well-being of the fetus, cardiotocography and ultrasound examination with the amniotic fluid index should be performed twice a week and the Doppler ultrasound once a week. General ultrasound controls could improve the detection of peripartum complications, but they did not increase the survival rates of neonates [72].

Previable PPROM pregnancies are associated with an extremely high risk of neonate complications. While specialist consultations are needed, studies have shown that pregnancies monitored in an outpatient could have better outcomes.

### 2.7. Neonatal Outcomes after Previable PPROM

PPROM-induced preterm birth is the leading cause (>95%) of neonatal morbidity and mortality [2,73,74]. Delivery between 22 and 28 weeks of gestation leads to an even higher risk of inappropriate lung maturation and impaired lung functioning and hence a low chance of survival of the fetus. Neonatal respiratory disorders (affecting 93% of newborns with birth weight <1500 g) [75,76] may be associated with irreversible changes in the lung tissue, called bronchopulmonary dysplasia (affecting up to 30% of mechanically ventilated newborns) [77,78]. Preterm birth is also associated with immaturity of the cardiovascular system [75], which further decreases the survival chances of the fetus.

Late-onset sepsis is another common complication. In one study by Stoll et al. conducted on a group of 6215 newborns, it was shown that up to 21% of premature children showed symptoms of bacteriemia [79]. Infection in premature newborns can have a sudden onset and escalate rapidly, quickly leading to death. Intraventricular bleeding affects up to 15% of very premature children, i.e., born <28 weeks of gestation with very low birth weight [80,81]. Periventricular hemorrhage and periventricular leukomalacia are often co-occurring characteristics of brain damage, and neuronal defects manifest as cerebral palsy in ca. 80% of children delivered <28 weeks of gestation [82]. According to available data, cerebral palsy is the most common cause of motor disability in children [83]. The occurrence of previable PPROM additionally increases the risk of nervous system damage [84].

Perinatal mortality is defined as stillbirth or death of a live-born child up to 7 completed days of life [2,85]. Early-neonatal death is defined as death between birth and the 7th day of life and late-neonatal death as death when occurring between the 7th and 28th day after delivery [2,74]. All the preterm labor complications discussed above might result in early and late-perinatal death. The duration of pregnancy and the birth weight of newborns are proportionated with the survival rate of neonates [86]. In the best case, newborns born before 28 weeks of gestation reach a survival rate of 60% [27,28,87].

In the study of Esteves et al., 80% of infants born from pregnancies complicated with PPROM between 18–26 weeks of gestation experienced severe adverse outcomes, with a 54% perinatal mortality. Only 18% of children had no postnatal complications. Antibiotic prophylaxis, gestational age, and birth weight were significant predictors of children’s survival [86,87]. In previable PPROM pregnancies after prophylactic administration of antibiotics and antenatal steroid therapy, fetal mortality was 17% for delivery between 24–27 weeks of gestation [88]. The study of Wagner et al. showed that 50% of children with PPROM at 24 weeks of gestation could be discharged from the hospital without major complications [89]. Neurodevelopment of children at 24 months after delivery after 7-day PPROM (at 24–31 weeks of gestation) was similar to the development of infants delivered at the same gestational age without PPROM [17].

## 3. Future Opportunities and Directions

To date, there is still no successful treatment method of previable PPROM. Nevertheless, the increasingly evolving technology may allow us to modify the management of pregnancies complicated with previable PPROM [90]. For example, the artificial intelligence methods based on algorithms created from individual risk factors can substantially improve the prediction of the survival rate of children with previable PPROM as well as the incidence of intrauterine infections and preterm labor [91]. The possible target for future opportunities is thus identifying pregnancies with a higher chance of survival and providing appropriate care of these pregnancies. Pregnancy termination could be recommended after an interdisciplinary assessment and evaluation with artificial intelligence for women with a low chance of delivering a living infant, depending on maternal well-being and the local law. Still, in those pregnancies, all prophylactic measures should be explained. Another possibility to improve neonatal outcomes could be an artificial placenta, which could manage the effective treatment of respiratory failure in premature infants [92]. Further research is necessary to develop appropriate methods and guidelines.

Summary of findings is presented in Table 3.

## 4. Conclusions

Prematurity remains a global health problem and is the cause of almost all neonatal complications, neonatal mortality, and irreversible neurological deficits. Clinicians must decide whether the risk of complications resulting from previable PPROM outweighs the risk of intrauterine infection. Further studies are needed to determine the gestation age cut-off point for continuing the pregnancy, which would be optimal for decreasing peripartum morbidity and mortality, and to identify factors that improve the prognosis for fetuses, e.g., the use of prophylactic antibiotic therapy in pregnancy. The gestational age of 32–34 seems reasonable to electively deliver pregnancies complicated by previable PROM to avoid developing the intrauterine infection. Previable PPROM remains a challenge because of the difficulties in diagnosis and its complications. Patients diagnosed with PPROM should remain under the comprehensive care of a perinatology center, with the monitoring of cardiotocography, ultrasound, laboratory tests, and vital signs. Nevertheless, outpatient monitoring is also possible and, according to some studies, leads to better results. Administration of steroids is usually needed, but the decision to start treatment should be made on a case-by-case basis. Additionally, antibiotic prophylaxis is shown to prevent intrauterine infections. Treatment with erythromycin with or without a combination of ampicillin alone or amoxicillin with erythromycin until delivery or within 7 days after PPROM seems to be the "gold standard" of prophylaxis of intrauterine infection and preterm delivery. Macrolides, such as clarithromycin and azithromycin, were found to be more effective and better tolerated by the patients. However, the initiation of treatment and dosage and the length of prophylactic antibiotic therapy should be analyzed in further prospective and randomized research. Artificial intelligence methods based on algorithms created from individual risk factors can substantially improve the prediction of the survival rate of children with previable PPROM as well as the incidence of intrauterine infections and preterm labor.

## Figures and Tables

**Table 1 diagnostics-12-02025-t001:** Summary of postulated premature rupture of membranes risk factors.

PPROM ^1^ Risk Factors
History of PPROM
History of preterm birth
Nulliparity
GDM ^2^ and insulin intake
Multiple pregnancies
Infections (vaginal/urinary)
Bleeding in the 1st trimester
BMI ^3^ < 18.5 kg/m^2^
Cigarette smoking
Low level of education

^1^ Preterm premature rupture of membranes; ^2^ gestational diabetes mellitus; ^3^ body mass index.

**Table 2 diagnostics-12-02025-t002:** Comparison of recommendations of selected scientific societies regarding antibiotic therapy in PPROM ^1^.

Authorship	Year	Antibiotics in the Management of PPROM *This Section Applies Only to Antibiotic Use in PPROM, Excluding Aspects Such As GBS ^2^ Prophylaxis or Treatment of an Intraamniotic Infection*
American College of Obstetriciansand Gynecologists [30]	2020	Pregnancy periviable (Less than 23–24 GWs^3^)—antibiotics may be considered as early as 20 0/7 GWs. Preterm pregnancy (24 0/7–33 6/7 GWs)—antibiotics recommended if no contraindications. A 7-day course of therapy—ampicillin and erythromycin i.v. ^4^ followed by amoxicillin and erythromycin p.o. ^5^ Substitution of erythromycin with azithromycin when not available or tolerated may be an alternative. Amoxicillin with clavulanic acid should not be used. Late preterm pregnancy (34 0/7–36 6/7 GWs)—GBS prophylaxis + treatment of possible intraamniotic infection and proceed toward delivery.
Royal College of Obstetricians and Gynecologists [31]	2019	Erythromycin should be given for 10 days following the diagnosis of PPROM or until labor; dosing schedule unclear, i.e., erythromycin 250 mg 4 times daily p.o. ^4^ Penicillin may be used in women who do not tolerate erythromycin. Amoxicillin with clavulanic acid should not be used.
National Institute for Health and Care Excellence [32]	2019	Erythromycin 250 mg 4 times a day p.o. for 10 days or until labor. Penicillin may be used in women who have contraindications or do not tolerate erythromycin—for a maximum of 10 days. Amoxicillin with clavulanic acid should not be used.
The Society of Obstetricians and Gynaecologists of Canada [33]	2017	Following PPROM at 32 GWs—antibiotics should be administered to women who are not in labor. PPROM at >32 GWs—administration of antibiotics to prolong pregnancy is recommended if fetal lung maturity cannot be proven and/or delivery is not planned. The following two regimens may be used: ∘ Ampicillin 2 g i.v. ^5^ every 6 h and erythromycin 250 mg i.v. every 6 h for 48 h, followed by amoxicillin 250 mg p.o. every 8 h and erythromycin 333 mg p.o. every 8 h for 5 days; ∘ Erythromycin 250 mg p.o. every 6 h for 10 days. Amoxicillin with clavulanic acid should not be used.

^1^ Preterm Premature Rupture of Membranes; ^2^ group B Streptococcus; ^3^ gestational weeks; ^4^ intravenous; ^5^ orally every eight hours for the next five days [15]. Even though steroids were not administered, the therapy with antibiotics alone reduced the rate of respiratory failure, necrotizing enterocolitis, and death of fetuses and newborns. Based on this study, the American Society of Gynecologists and Obstetricians (ACOG) recommends using the above-mentioned antibiotic prophylaxis for seven days in the case of PROM between 24 and 34 weeks of pregnancy [30].

**Table 3 diagnostics-12-02025-t003:** Summary of findings.

PPROM
**Definition**: PPROM is a medical incident involving the loss of integrity of the amniotic membranes and amniotic fluid before the beginning of uterine contractions before 37 gestational weeks **Prevalence**: around 3% of pregnancies **Risk factors**: history of PPROM, history of preterm birth, nulliparity, GDM and insulin intake, multiple pregnancies, infections (vaginal/urinary), bleeding in the 1st trimester, BMI <18.5 kg/m^2^, cigarette smoking, lower level of education **Prognosis**: most patients deliver within 7–14 days from PPROM **The greatest threat**: intrauterine infection and preterm birth
**Management**: consider antibiotics, tocolysis, and steroids while taking into account gestational age and fetus prognosis as well as clinical image; closely monitor the condition of the fetus and the mother (among others: CTG, ultrasound, and laboratory and physical parameters)
**The greatest challenge and future studies direction**: finding the optimal time for delivery and balancing the benefits and risks ratio for both the fetus and the mother

BMI, body mass index; CTG, cardiotocography; GDM, gestational diabetes mellitus; PPROM, premature rupture of membranes.

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
