# Peer review of "The Management of Pregnancy Complicated with the Previable Preterm and Preterm Premature Rupture of the Membranes: What about a Limit of Neonatal Viability?—A Review"

_diagnostics, 2022, doi:10.3390/diagnostics12082025_

Round 1

Reviewer 1 Report

In this review authors reviewed the available data regarding the prevention of preterm delivery caused by PPROM and evaluated the possible therapeutic options.

This manuscript is clear and generally well written but some points need to be improved. In particular: 

Introduction: Authors need to point out that also chorioamnionitis are involved in PROM since inflammation found in this pathology break down the tight junctions of placental membranes (PMID: 26739007). 

It would be useful adding a summary table at the end of each chapter summarizing the studies analised by the authors. It would make the manuscript way more easy to read and understand. 

11. Conclusions: It would be interesting a short note regarding the role of natural compounds such as Vitamin C and Curcumin in preventing premature delivery since both these compounds showed beneficial effects in preventing this pregnancy complication (PMID: 23682322 and 33477354). This point is very useful and interesting since both these compounds could be used alone or in combination with antibiotics or  steroids to improve the outcome of these pregnancies. 

Author Response

This manuscript is clear and generally well written but some points need to be improved. In particular: 

Introduction: Authors need to point out that also chorioamnionitis are involved in PROM since inflammation found in this pathology break down the tight junctions of placental membranes (PMID: 26739007). 

- Thank you very much for your remark. We meant an intrauterine infection as a general term, including chorioamnionitis. But you are undoubtedly right that it should be mentioned in this place.

It would be useful adding a summary table at the end of each chapter summarizing the studies analised by the authors. It would make the manuscript way more easy to read and understand. 

- You are probably right that such a summary could be helpful and make the study clearer, but there would be a lot of tables, and it could hide the actual discussion about the aimed problem in our study.  

  1. Conclusions: It would be interesting a short note regarding the role of natural compounds such asVitamin C and Curcumin in preventing premature delivery since both these compounds showed beneficial effects in preventing this pregnancy complication (PMID: 23682322 and 33477354). This point is very useful and interesting since both these compounds could be used alone or in combination with antibiotics or steroids to improve the outcome of these pregnancies. 

- Thank you for this information. Appropriate changes were provided.

Reviewer 2 Report

I was excited to read this paper reviewing the diagnosis and management of preterm prelabour rupture of membranes at the limit of neonatal viability. Unfortunately, this paper does this in name (Title) only...and does NOT provide readers any information about management of previable PPROM at all, and sadly misses many papers about previable PPROM published in the literature.

Overall, the sections of the review seem out of order and do not flow logically from diagnosis to management to pregnancy issues and lastly neonatal/longer-term outcomes in the offspring.....and there is almost nothing discussed about diagnosis at all, almost nothing discussed about management that pertains to PPROM at the limits of viability (rather just presents info for management of 'regular' PPROM)....and the bulk of the paper is written about outcomes.

TITLE:

1. The title is MISLEADING as it suggests this paper is a review about PPROM at the limits of viability.....when it is just another paper review about 'regular' PPROM. There is also very little offered by the authors regarding diagnosis at all....and much of the paper is about outcomes and complications.

INTRODUCTION.

1. Last line of paragraph 2 is awkward as 22wk is NOT universally considered as the definition for miscarriage (irrespective of presence/absence of PPROM) --> would recommend using the WHO definitions and differentiating stillbirth from miscarriage vs previable birth. At any rate, this sentence is misleading because given the latency between early PPROM and delivery, many cases of PPROM at the limits of viability do NOT necessarily delivery early.

2. The aim of the review as stated in the last paragraph of the Intro is NOT achieved by this paper: at best, this paper aims to review the risk factors and outcomes after 'regular' PPROM and provides options for management of 'regular' PPROM.

3. In many places, the definition of viability is also changing to 22 or 23 weeks...so the definition provided by the authors of 24 weeks seems outdated if this is meant to be a modern review of this clinical complication.

Pregnancy after PPROM

1. This section should be renamed to perinatal outcomes, as only a minority of pregnancy issues are discussed here (vs scattered throughout the paper)

2. This section seems out of place as it mainly covers neonatal outcomes of PPROM....and doesn't address the obstetrical issues that precede it (or the MAJOR issues of latency and how they correspond to timing or gestational age of delivery....which itself is a main determinant of neonatal outcomes from PPROM). Latency should be

3. Any paper claiming to be about PPROM at the limits of viability must also address -- issues of pulmonary hypoplasia and other morbidity SPECIFIC to early PPROM (i.e. critical period of fetal lung development embryologically) in a less generic way, which this paper fails to do.

ANTIBIOTICS

1. There is no information provided about management of previable PPROM - only a reiteration of management and supporting studies for management of or 'regular' PPROM (for which the published literature is already robust)

TOCOLYSIS

1. This section (and definition of tocolysis itself) is poorly worded and needs a major overall.

2. Magnesium sulphate is almost universally refuted as an effective tocolytic agent, and nothing mentioned in the last paragraph of this section is specific to PPROM.

STEROIDS:

1. Any suggestion that steroid administration should be used with caution because of risks of intrauterine infection in the setting of PPROM is dangerous, and has been universally refuted by modern neonatal medicine/literature/practices. This is an important component in the care of patients with PPROM and requires a MAJOR overhaul in writing to better reflect the current state of medical practice.

PREGNANCY LATENCY

1. This section needs to appear much earlier in the review

FUTURE OPPORTUNITIES

1. This should appear at the end of the review

CHANGES IN NEONATES

1. This section doesn't make sense - it it about neonatal outcomes....or previable delivery.....or management and associated latency? Very confusion.

It is written generically and represents little value to the topic area.

CONCLUSIONS

1. NOTHING is added to the conversation or understanding about management of PPROM at the limits of viability. Where are the references to the growing number of studies about previable pPPROM?

Overall, this paper misses the mark and misleads the reader into thinking this review is about something with an existing gap in knowledge....when it is just an average review about 'regular' PPROM.

Author Response

Overall, the sections of the review seem out of order and do not flow logically from diagnosis to management to pregnancy issues and lastly neonatal/longer-term outcomes in the offspring.....and there is almost nothing discussed about diagnosis at all, almost nothing discussed about management that pertains to PPROM at the limits of viability (rather just presents info for management of 'regular' PPROM)....and the bulk of the paper is written about outcomes.

TITLE:

  1. The title is MISLEADING as it suggests this paper is a review about PPROM at the limits of viability.....when it is just another paper review about 'regular' PPROM. There is also very little offered by the authors regarding diagnosis at all....and much of the paper is about outcomes and complications.

- Thank you for your remark. We aimed to assess both regular PPROM and PPROM at the limits of viability. Unproportional information included in the review with a more significant part about regular PPROM is performed because of very little information about PPROM before 22 – 24 wk. Additional ongoing trials were discussed in the study, according to Your remark.

- You are also right about little information about diagnostics. We are supposed to concentrate in this article on PPROM management as it is more complicated and important. Moreover, discussed diagnostics methods, when used properly, can diagnose a significant count of PPROMs. Nevertheless, specific studies to identify PPROM at the limits of viability should be performed.  

- The Title was changed according to your suggestion.

INTRODUCTION.

  1. Last line of paragraph 2 is awkward as 22wk is NOT universally considered as the definition for miscarriage (irrespective of presence/absence of PPROM) --> would recommend using the WHO definitions and differentiating stillbirth from miscarriage vs previable birth. At any rate, this sentence is misleading because given the latency between early PPROM and delivery, many cases of PPROM at the limits of viability do NOT necessarily delivery early.

- You are correct that “22wk is NOT universally considered as the definition for miscarriage”. There are various definitions of preterm labour and miscarriage, and the line is between the 22nd and 24th weeks of gestation. We took the most restricted cut-off point to include and assess all studies about PPROM. We provided appropriate changes in this area of our manuscript.

  1. The aim of the review as stated in the last paragraph of the Intro is NOT achieved by this paper: at best, this paper aims to review the risk factors and outcomes after 'regular' PPROM and provides options for management of 'regular' PPROM.

- Thank you for your remark. We aimed to assess both regular PPROM and PPROM at the limits of viability. Unproportional information included in the review with a more significant part about regular PPROM is performed because of very little information about PPROM before 22 – 24 wk. Additional ongoing trials were discussed in the study, according to Your remark.

  1. In many places, the definition of viability is also changing to 22 or 23 weeks...so the definition provided by the authors of 24 weeks seems outdated if this is meant to be a modern review of this clinical complication.

- Thank you for this remark. There are various definitions of preterm labour and miscarriage, and the line is between the 22nd and 24th weeks of gestation. We took the most restricted cut-off point of the 22nd week of gestation to include and assess all studies about PPROM. We provided appropriate changes in this area of our manuscript.

Pregnancy after PPROM

  1. This section should be renamed to perinatal outcomes, as only a minority of pregnancy issues are discussed here (vs scattered throughout the paper)

- Thank you for this remark. The section “Pregnancy after PPROM: was removed, and the information was moved to the “Pregnancy latency” section.

  1. This section seems out of place as it mainly covers neonatal outcomes of PPROM....and doesn't address the obstetrical issues that precede it (or the MAJOR issues of latency and how they correspond to timing or gestational age of delivery....which itself is a main determinant of neonatal outcomes from PPROM). Latency should be

- Thank you for this remark. The appropriate changes were added, and the information was moved to the “Pregnancy latency” section.

  1. Any paper claiming to be about PPROM at the limits of viability must also address -- issues of pulmonary hypoplasia and other morbidity SPECIFIC to early PPROM (i.e. critical period of fetal lung development embryologically) in a less generic way, which this paper fails to do.

- Thank you for this remark. The appropriate changes were added, and the information was moved to the “Pregnancy latency” section.

ANTIBIOTICS

  1. There is no information provided about management of previable PPROM - only a reiteration of management and supporting studies for management of or 'regular' PPROM (for which the published literature is already robust)

- Thank You for your remark. The most recent clinical trial was added to the review.

TOCOLYSIS

  1. This section (and definition of tocolysis itself) is poorly worded and needs a major overall.

- Thank you for this remark. The more specific PPROM information was added. Nevertheless, as mentioned in the manuscript, the use of tocolysis is very constricted.

  1. Magnesium sulphate is almost universally refuted as an effective tocolytic agent, and nothing mentioned in the last paragraph of this section is specific to PPROM.

- Thank you for this remark. The correction of the sentence was written.

STEROIDS:

  1. Any suggestion that steroid administration should be used with caution because of risks of intrauterine infection in the setting of PPROM is dangerous, and has been universally refuted by modern neonatal medicine/literature/practices. This is an important component in the care of patients with PPROM and requires a MAJOR overhaul in writing to better reflect the current state of medical practice.

- You are undoubtedly right. We did not suppose to take even into consideration the fact that steroids are necessary as an RDS prophylactic. Only side effects should be mentioned as a result of any treatment or prophylactic intervention. Appropriate changes were provided to clarify this part of the manuscript.

PREGNANCY LATENCY

  1. This section needs to appear much earlier in the review

- You are right. The section was moved.

FUTURE OPPORTUNITIES

  1. This should appear at the end of the review

- You are right. The section was moved.

CHANGES IN NEONATES

  1. This section doesn't make sense - it it about neonatal outcomes....or previable delivery.....or management and associated latency? Very confusion.

It is written generically and represents little value to the topic area.

- The title of the sections was changed, and more specific information about PPROM was added. Nevertheless, the consequences of preterm birth are very similar to other reasons of preterm delivery.

CONCLUSIONS

  1. NOTHING is added to the conversation or understanding about management of PPROM at the limits of viability. Where are the references to the growing number of studies about previable pPPROM?

- Thank you for your comment. We are supposed to summarise the published knowledge about what is known about pPPROM and regular PPROM.

Overall, this paper misses the mark and misleads the reader into thinking this review is about something with an existing gap in knowledge....when it is just an average review about 'regular' PPROM.

- Thank you for your review. We supposed to assess both regular PPROM and PPROM at the limits of viability. Unproportional information included in the review with a bigger part about regular PPROM is performed because of very little information about PPROM before 22 – 24 wk. Additional ongoing trials were discussed in the study, according to Your remark.

Reviewer 3 Report

The diagnostics and management of the pregnancy complicated with the preterm prelabour rupture of membranes at the limit of neonatal viability – a review, is a piece of work, authors trying to summarise the PPROM and neonatal outcome.

These are my suggestions

1.      Line 88-89: Neurological  deficits observed in a newborn or diagnosed at a later stage of development do not progress during the child’s development and tend to remain constant. The sentence is incorrect. Many Neurological  deficits progress as the as progresses.

2.      Line 93: definitions of perinatal mortality need to be corrected.

perinatal mortality is the sum of still birth (fetal death) and early neonatal death (ENND) i.e. death of live newborn before the age of 7 completed days (https://bmcpregnancychildbirth.biomedcentral.com/articles/10.1186/s12884-017-1420-7; WHO definition, https://en.wikipedia.org/wiki/Perinatal_mortality#cite_note-2 )

3.      Line: 181,82: The most common pathogens isolated from a vaginal area of women with PROM include Streptococcus spp., Staphylococcus aureus, Escherichia coli, Proteus mirabilis,Bacteroides spp. and Klebsiella pneumonia, Provide references.

4.      Line 237-39 : The use of  prenatal steroids may be associated with maternal side effects and result in generalised bacterial infection, tuberculosis, viral infections (shingles, Herpes), uncontrolled metabolic diseases, including diabetes, unresponsive hypertension, or active gastric ulcer and duodenum. Need to be supported.

5.      Line 256: A Cochrane review from 2014 did not find enough evidence to establish the superiority of hospital and home care of the patients with PPROM. In developing nations and underdeveloped countries it is always better have the patients monitored in hospital rather than home care. Justify that home care is equal as hospital setup as per your view. How to take care the neonates in home care of the patients with PPROM?

6.      Preterm labour negatively impacts not only the physical, but also the mental health 322 of women in the perinatal period. As a result of sudden changes in the concentration of 323 steroid hormones in a woman’s body, after physiological pregnancy and physiological 324 delivery, mood disorders called baby blues are often diagnosed[92]. Separation of the 325 mother from the newborn hospitalisation of a newborn child in the intensive care unit due 326 to complications of prematurity and associated fear for the baby’s life may further exac- 327 erbate these mood disorders, even leading to postpartum depression[93,94]. Patients with 328 a PPROM feel anxiety associated with the risk of preterm labour, which predisposes them 329 to depression during pregnancy and postpartum[95].

The above paragraph is not a future opportunities. Should be taken out.

Rather, biglycan and decorin, together with serum protein sex hormone-binding globulin (SHBG) as a promising second-trimester prenatal serum screening-based biochemical model with an ability to predict pPROM in asymptomatic women. (https://www.ncbi.nlm.nih.gov/pmc/articles/PMC8260062/)

Besides, insulin-like growth factor binding protein-1 (IGFBP-1) and placental alpha macroglobulin 1 (PAMG-1) are other marker to detect pPROM in asymptomatic women.

7.       10. Changes in neonates may changes as Neonatal outcome in pPROM, and 

Discuss the points accordingly.

General comments

8.      Conclusion made is general, they should focus the extra thing they have made out of this review, which is not clear.

9.      You should provide a summery table to overlook your findings from literature and specific suggestions to improve the diagnosis or treatment for clinicians

10.  Few places there is repetitions of facts eg. They all warn against the use of amoxicillin with clavulanic acid due to the proven increased risk of  neonatal necrotising enterocolitis[2,45–47]. Line 123-24 & line 118-19

11.  This presentation is like a minireview only.

Author Response

  1. Line 88-89: Neurological deficits observed in a newborn or diagnosed at a later stage of development do not progress during the child’s development and tend to remain constant. The sentence is incorrect. Many Neurological deficits progress as the as progresses.

- Thank you for your remark. The appropriate changes were made.

  1. Line 93: definitions of perinatal mortality need to be corrected. 

perinatal mortality is the sum of stillbirth (fetal death) and early neonatal death (ENND) i.e. death of live newborn before the age of 7 completed days(https://bmcpregnancychildbirth.biomedcentral.com/articles/10.1186/s12884-017-1420-7; WHO definition, https://en.wikipedia.org/wiki/Perinatal_mortality#cite_note-2 )

- I am sorry for this mistake. Appropriate changes were provided.

  1. Line: 181,82: The most common pathogens isolated from a vaginal area of women with PROM include Streptococcus spp., Staphylococcus aureus, Escherichia coli, Proteus mirabilis, Bacteroides spp. and Klebsiella pneumonia, Provide references.

- Thank you, references were added.

  1. Line 237-39: The use of prenatal steroids may be associated with maternal side effects and result in generalised bacterial infection, tuberculosis, viral infections (shingles, Herpes), uncontrolled metabolic diseases, including diabetes, unresponsive hypertension, or active gastric ulcer and duodenum. Need to be supported.

- Thank you, references were added.

  1. Line 256: A Cochrane review from 2014 did not find enough evidence to establish the superiority of hospital and home care of the patients with PPROM. In developing nations and underdeveloped countries it is always better have the patients monitored in hospital rather than home care. Justify that home care is equal as hospital setup as per your view. How to take care the neonates in home care of the patients with PPROM?

- Thank you for your remark. This part was probably not written clearly enough, and the clarification was added to the manuscript.

- The idea of outpatient clinic care is to minimalise the risk of intrauterine infection. Women discharged from hospitalisation need further control in an outpatient clinic, but the place of resistance is their own house. According to that, home bacterial flora surrounds them. Because of it, the risk of intrauterine infections is minimalised.

- Postnatal care is impossible after preterm delivery. Therefore, hospital labour is required. 

  1. Preterm labour negatively impacts not only the physical, but also the mental health of women in the perinatal period. As a result of sudden changes in the concentration of steroid hormones in a woman’s body, after physiological pregnancy and physiological delivery, mood disorders called baby blues are often diagnosed[92]. Separation of the mother from the newborn hospitalisation of a newborn child in the intensive care unit due to complications of prematurity and associated fear for the baby’s life may further exacerbate these mood disorders, even leading to postpartum depression[93,94]. Patients with a PPROM feel anxiety associated with the risk of preterm labour, which predisposes them to depression during pregnancy and postpartum[95].

The above paragraph is not a future opportunities. Should be taken out.

- The idea of adding this paragraph was to show the complex impact of PPROM, but You are probably right. The psychological aspect of the whole section was removed.

Rather, biglycan and decorin, together with serum protein sex hormone-binding globulin (SHBG) as a promising second-trimester prenatal serum screening-based biochemical model with an ability to predict pPROM in asymptomatic women. (https://www.ncbi.nlm.nih.gov/pmc/articles/PMC8260062/)

Besides, insulin-like growth factor binding protein-1 (IGFBP-1) and placental alpha macroglobulin 1 (PAMG-1) are other marker to detect pPROM in asymptomatic women.

- Thank You for this remark. We decided to change the Title according to the little information present about diagnostics of PPROM. According to this change, we would rather not including additional information about diagnosis of PPROM.

  1. 10. Changes in neonates may changes asNeonatal outcome in pPROM, and  

Discuss the points accordingly.

- Thank you, this correction was provided.

General comments

  1. Conclusion made is general, they should focus the extra thing they have made out of this review, which is not clear.

- A more specific conclusion to PPROM at the limit of neonatal viability was added.

  1. You should provide a summery table to overlook your findings from literature and specific suggestions to improve the diagnosis or treatment for clinicians

- The summary table 3. was added to the manuscript.

  1. Few places there is repetitions of facts eg. They all warn against the use of amoxicillin with clavulanic acid due to the proven increased risk of neonatal necrotising enterocolitis[2,45–47]. Line 123-24 & line 118-19

- You are right about these repetitions, but they are needed because of describing different studies with similar findings.

Round 2

Reviewer 3 Report

Well revised

Author Response

Thank you very much for your comments and the review.